# Risk of Dependence on Sport in Relation to Body Dissatisfaction and Motivation

**Inmaculada Tornero-Quiñones, Jesús Sáez-Padilla \***[ID]**, Estefanía Castillo Viera, Juan José García Ferrete and Ángela Sierra Robles**

Faculty of Education, Psychology and Sports Sciences, University of Huelva, 21071 Huelva, Spain; inmaculada.tornero@dempc.uhu.es (I.T.-Q.); estefania.castillo@dempc.uhu.es (E.C.V.); ferrete_@hotmail.com (J.J.G.F.); sierras@uhu.es (Á.S.R.)
**\*** Correspondence: jesus.saez@dempc.uhu.es; Tel.: +34-679096680

**Abstract:** The aim of this study was to investigate the risk of dependency on physical exercise in individual sportspeople and the relationship with body dissatisfaction and motivation. Two hundred and twenty-five triathletes, swimmers, cyclists, and athletics competitors aged 18 to 63 years old took part in the study, of which 145 were men (M = 35.57 ± 10.46 years) and 80 were women (M = 32.83 ± 10.31 years). The EDS-R (Exercise Dependence Scale-Revised) was used to study dependency on exercising, the Body Shape Questionnaire (BSQ) was used to study body dissatisfaction, the Behaviour Regulation in Exercise Questionnaire (BREQ-3) was used to determine the participants' motivation and the BIAQ was used to analyse conducts of avoidance to body image. The obtained results show that 8.5% of the subjects have a risk of dependency on exercise and 18.2% tend to have corporal dissatisfaction, without meaningful differences in the kind of sport they practiced. However, there were important differences concerning the dependency on physical exercise (15% vs. 4.8%) and body dissatisfaction (31.1% vs. 11%) in relation to sex, with the higher percentages referring to women. Introjected regulation and the conduct of food restriction were predictor variables of the dependency on exercise and corporal dissatisfaction. Also, the number and duration of sessions; the age of the participants; the integrated, introjected, and external regulations surrounding social activities; and eating restraints could all predict dependence on physical exercise (DPE).

**Keywords:** addiction; triathletes; body image; behaviour regulation

## 1. Introduction

Nowadays, the multitude of benefits associated with participating in a sport or physical exercise on a regular basis are known on physical, psychological, aesthetic and social levels [1–4]. However, in recent decades, some new research has emerged on the addiction that sport can create, having negative consequences not only at physical or physiological levels (e.g., abstinence, tolerance) but also psychological (e.g., anxiety, depression) or behavioural (e.g., reduction of other activities) levels [5–7].

Initially, the term "addiction" or the concept of dependence in relation to physical exercise was not given negative conjectures. Glasser [8] referred to exercise as a positive addiction, in relation to the health benefits it has, stating that exercise addiction is beneficial. However, Morgan [9] studied and recognized the negative effects that could appear with sport addiction such as injuries, over-training, social isolation, and psychological problems.

Subsequently, Ogden, Veale and Summers [10] defined the dependence on physical exercise (DPE, hereafter) as a combination of biomedical characteristics similar to those of addictions, such as withdrawal symptoms and stereotyped behaviours in addition to other psychosocial aspects such as

interference with social/family life and positive gratifications. Research along this line, such as that of Sussman, Lisha, and Griffiths [11] has found similarities between sports addiction and drug addiction.

Addiction to sport is manifested by its influence and the abuse of it in day-to-day life, causing other factors of life to be ignored. Whiting [12] concluded that the non-practice of sports on some days generated anxiety, abstinence, as well as physical and moral discomfort in those athletes with symptoms or risk of DPE.

Several studies have associated DPE with eating disorders and body disorder or dissatisfaction (BI, hereafter) [7,13,14] in relation to the perceived importance of greater body care or maintaining a good physical appearance, sometimes even exceeding the limits, with athletes developing habits, behaviours, or thoughts that distort their body image [15].

An important aspect to take into account in sports practice is motivation and its relationship with the theory of self-determination [16]. This theory describes how much motivation exists and what kind of motivation leads us to practice sports. This motivation may be related to the DPE, either as a predecessor or as a consequence thereof [17,18].

Sex or age, as well as the sport practiced or the body mass index (BMI, hereafter), are factors that are related to DEP, BI, and motivation [19–21]. More and more studies are being conducted on addiction to sport and what factor or factors it may be related to. In fitness centres [17], it has been shown that 7% of members could be considered at risk of exercise dependence. Regarding individual sports, Ruiz-Juan and Zarauz [22] studied Spanish marathoners, who showed a medium-high level of addiction to training. In investigations on triathletes, Blaydon and Lindner [13] and Youngman [23] concluded that 30.4% and 19.9% of study participants, respectively, were likely to have sports addiction or DPE. Latorre et al. [24] found that the prevalence of DPE in different endurance sports (triathlon, swimming, cycling and athletics) was 13.6%, with triathletes having the highest rate of DPE.

On the other hand, in relation to sex, the results are not very clear, since Guszkowska and Rudnicki [25] found that males were more addicted to sport than females; however, Youngman [23] concluded that girls had the highest rate of DPE, although the difference between genders was not significant.

Nevertheless, despite what we know today, Forte and Ferreira and Ruiz-Juan and Zarauz [26,27] stated there are still many aspects to learn more about sports addiction or DPE, as well as which sports carry higher risks of DPE and BI and how these variables interweave with personal and motivational differences. On the other hand, the studies carried out usually use students or gym clients as participants, and DPE has rarely been investigated, for example, in outdoor sports.

Therefore, based on the reviewed studies, this research aims to (a) compare the current state of triathlon and the sports that compose it (swimming, cycling, and running); (b) to determine and study the differences among DPE, BI, and the motivations of individual sport athletes according to modality and sex; (c) to establish the relationship between DPE and BI together with motivation, avoidance of BI, age, BMI, and the duration and number of sessions completed.

## 2. Materials and Methods

### 2.1. Participants

Two hundred and twenty-five athletes aged between 18 and 63 years participated in the study, of whom 145 were men (M = 35.57 ± 10.46 years) and 80 were women (M = 32.83 ± 10.31 years). The selected sample consisted of swimmers, cyclists, background athletes, and triathletes at a competitive amateur level who were classified at the Andalusian level. The parameters of inclusion in the study were being of legal age, performing at least 3 training sessions per week, and being active at this time.

### 2.2. Instruments

The Spanish version of the "Exercise Dependence Scale-Revised" (EDS-R) [28] produced by Sicilia and González-Cutre [29] was used. The scale is composed of 21 items that are used to obtain a global score on exercise dependence (as this score increases, there is a greater risk of dependence) and a score for each of the seven symptoms that define it. The questionnaire is headed by the statement "In the fitness centre …" and the responses are presented in the Likert format from 1 (never) to 6 (always). The subscales are abstinence (e.g., I practice physical exercise to avoid feeling in bad humour), continuation (e.g., I practice exercise despite repeated physical problems), tolerance (e.g., I constantly increase the intensity of my physical practice to achieve the desired benefits or effects), lack of control (e.g., I am unable to reduce the total time that I practice physical exercise), reduction of other activities (e.g., I would like to practice more physical exercise rather than being with my family and friends), time (e.g., I dedicate a significant amount of time to practicing physical exercise), time (e.g., I spend a lot of time on physical activity), and desired effects (e.g., I practice physical exercise for longer than I usually want to). In this study, an internal consistency coefficient (Cronbach's alpha) of 0.92 was obtained.

The scale also allows the practitioners of physical exercise to be classified into three groups: At risk of dependence (score of 5–6 in at least three of the seven criteria), symptomatic non-dependent (score of 3–4 in a minimum of three criteria or a score of 5–6 combined with scores of 3–4 in three criteria, but without meeting the conditions of being at risk), and asymptomatic non-dependent (score of 1–2 in at least three criteria, but without actually meeting the conditions of the symptomatic non-dependent condition).

The version of the Body Shape Questionnaire (BSQ) [30] adapted to the Spanish population by Raich et al. [31] was used for the analysis of body satisfaction. This is a questionnaire comprising 34 items that are evaluated by the following scale (1 = never, 2 = rarely, 3 = sometimes, 4 = often, 5 = very often, 6 = always), so the test range is 34–204. Following Cooper and Taylor (1988), from a total BSQ score of >80 indicates BI risk. In this study we obtained a Cronbach's alpha value of 0.96.

To analyse the motivation towards sport, we used the Spanish version of the Behaviour Regulation in Exercise Questionnaire (BREQ-3) [32] developed by González-Cutre, Sicilia and Fernández, [33], which is composed of 23 items divided into 6 factors. The questionnaire is headed by the statement "I do physical exercise …" and the answers are presented in a Likert format from 0 (not true) to 4 (totally true). The dimensions that make up the questionnaire are intrinsic regulation *(e.g., Because I think exercise is fun)*, integrated regulation *(e.g., Because it is in accordance with my way of life)*, identified regulation *(e.g., Because I value the benefits of physical exercise)*, introjected regulation *(e.g., "Because I feel guilty when I do not practice it")*, external regulation *(e.g., Because the others tell me I should do it)* and de-motivation *(e.g., I do not see why I have to do it)*. In this study we obtained an internal consistency coefficient (Cronbach's alpha) of 0.642 for the regulation of behaviour during exercise.

Finally, the BIAQ measures the behavioural aspect of BI [34]. It consists of 19 items, in which different behaviours of avoidance related to body appearance are investigated. The response scale is of the Likert type, with options from 1 (never) to 5 (always). The items are grouped into four subscales, which evaluate behaviours of avoidance related to (1) the method of wearing clothes (items 1, 2, 3, 4, 13, 15, 16, 17 and 18) *(e.g., I wear very loose clothes)*, (2) social activities (items 8, 9, 10 and 11) *(e.g., I do not go to social gatherings if this involves eating)*, (3) the restriction of food (items 5, 6, and 7) *(e.g., Fasting for a day or more)* (4) and a subscale with which we intended to measure checking behaviours (items 12, 14 and 19) *(e.g., I look at myself in the mirror)*. To interpret the scores, it must be taken into account that high scores indicate a greater frequency of behaviours of avoidance due to BI. In relation to the reliability of the scale, the Cronbach's alpha value was 0.79 for the total score on the scale in the original study.

## 2.3. Process

The participants signed an informed consent in which they were briefly informed of the type of study that was being carried out, were told that their answers were anonymous and confidential, that participation was voluntary and that the results would be available to them at the end of the investigation. The EDS-R, BSQ, BREQ-3, BIAQ and a data questionnaire referring to different sociodemographic variables were provided. The data were collected at different competitive events at the Andalusian level (Andalucía triathlon championships (medium and Olympic distance), Andalusian cycling circuit, open water swimming events, popular races).

## 2.4. Analysis of Data

First of all, the descriptive statistics (mean, standard deviation and percentages) of the sample are shown. The *t*-test for independent samples and the Chi-square test were used by means of a contingency table of categorical variables to analyse the homogeneity of groups in relation to sociodemographic characteristics. For the comparison of groups, analysis of variance (ANOVA) with post hoc testing was performed by means of Tukey adjustment for samples with equal variance and Games–Howell for those that had differences in variance, carrying out, in this case, a robust Welch analysis. In addition, the percentages of practitioners who could be considered to be asymptomatic non-dependent (AND), symptomatic non-dependent (SND), and at risk of dependence (RD), in the total sample and within each of the sports were calculated. Similarly, the sample was studied to determine the risk of BI. On the other hand, the analysis of variance with segmentation by sport and sex was conducted. We also used a Pearson correlation analysis between the variables and a multiple linear regression with DPE and BI as dependent variables. The level of significance was established to be $p < 0.05$. The statistical analysis of the data was performed through the statistical program SPSS., V.24.0 for Windows, (SPSS Inc., Chicago, USA).

## 3. Results

Table 1 describes the results obtained from the relationships among the different sociodemographic variables and the practiced sport modalities. It can be observed that there were significant differences in age, with swimmers being the youngest; in occupation, with triathletes more likely to have an occupation; the level of study, with triathletes being more likely to have studied at a higher level; years of training, with swimmers having the greatest percentage of individuals with more than 10 years of training, despite being the youngest group, followed by cyclists; and the condition of being federated, being greater in swimmers and triathletes than in the other two modalities. Swimmers and triathletes were found to complete more sessions, while cyclists were found to spend more time training in each session. Finally, swimmers were more likely to have a coach in relation to other sports.

Table 2 shows the results obtained from the EDS-R, BSQ, BREQ-3, and BIAQ questionnaires corresponding to each of the analysed sport modalities. In the EDS-R, the cyclists presented higher scores but without significant differences with respect to the rest of the sports. There were significant differences in the time of exercise between the triathletes and the athletics competitors and between the triathletes and the swimmers. There were no significant differences among the different sports modalities in the BSQ total scores. However, we did find significant differences in the BREQ-3 scores, specifically, regarding the higher intrinsic regulation in cyclists than in athletics competitors and even more so in cyclists in comparison to swimmers. Integrated regulation was found to be greater in triathletes than in swimmers. Identified regulation was greater in cyclists than in swimmers, and external regulation was higher in swimmers than in cyclists.

When sex was taken into account, there were significant differences (t (223) = 5.384, $p < 0.001$; d = 0.78) in the BSQ, with women obtaining higher scores (M = 71.16 ± 23.11) than men (M = 53.8 ± 21.35); in the BIAQ (t (223) = 3.790, $p < 0.001$; d = 0.46) where women also obtained higher scores (M = 1.92 ± 0.39) than men (M = 1.71 ± 0.51); and in the clothing mode dimension

($t$ (223) = 2.332, $p$ = 0.021; d = 0.34), where females obtained higher scores (M = 1.75 ± 0.51) than males (M = 1.59 ± 0.44). Significant differences were also found (t (223) = 7.022, $p$ < 0.01, d = 1.00) in BMI, with greater BMI values found in men (M = 23.19 ± 2.33) than in women (M = 21.04 ± 1.93).

In addition, the differences among sports by sex were observed through the results obtained in the EDS-R, BSQ, BREQ-3, and BIAQ. In the male sex, significant differences were found for intrinsic regulation (F (3, 141) = 3.886; $p$ < 0.01) where the cyclists obtained higher scores (M = 3.73 ± 0.36) than the swimmers (M = 3.37 ± 0.52). Similarly, for identified regulation (F (3, 141) = 2.729, $p$ = 0.029), the cyclists (M = 3.66 ± 0.49) obtained higher scores than the swimmers (M = 3.31 ± 0.55). Regarding exercise time (F (3, 141) = 4.272, $p$ < 0.01), triathletes (M = 11.85 ± 3.20) obtained higher scores than athletics competitors (M = 9.13 ± 0.65; $p$ = 0.012) and swimmers (M = 9.36 ± 0.61; $p$ = 0.017. Finally, significant differences were also found (F (3, 75.056) = 3.099, $p$ < 0.01) in the BMI values between triathletes (M = 22.71 ± 1.63) and swimmers (M = 24.36 ± 3.05; $p$ = 0.028), and the difference was even greater between cyclists (M = 22.66 ± 2.18) and swimmers ($p$ = 0.036). On the other hand, in the female gender, significant differences were only found for integrated regulation (F (3, 39.075) = 4.368, $p$ = 0.01), which was higher in cyclists (M = 3.89 ± 0.18) than in swimmers (M = 3.54 ± 0.57; $p$ = 0.044), and in demotivation (F (3, 38.267 = 3.915; $p$ = 0.016), which was higher in swimmers (M = 0.45 ± 0.59) than in cyclists (M = 0.06 ± 0.17; $p$ = 0.021). A significant difference was also found (F (3.76) = 2.954, $p$ = 0.038) in the practice time, with the score being higher in triathletes (M = 12.13 ± 3.58) than in athletics competitors (M = 8.41 ± 3.62; $p$ = 0.025).

**Table 1.** Sociodemographic characteristics in relation to the practiced sport modality.

| | | Swimming (n = 60) | Cycling (n = 56) | Athletics (n = 47) | Triathlon (n = 62) | *p*-value |
|---|---|---|---|---|---|---|
| Gender (%) | Males | 36 (60) | 40 (71.4) | 30 (63.8) | 39 (62.9) | 0.619 |
| | Females | 24 (40) | 16 (28.6) | 17 (36.2) | 23 (37.1) | |
| Age (SD) | | 29.18 (10.58) | 35.55 (11.49) | 38.04 (10.11) | 36.35 (7.49) | <0.001 |
| Employment (%) | Work | 30 (50) | 45 (80.4) | 35 (74.5) | 56 (90.3) | <0.001 |
| | Not work | 2 (3.3) | 1 (1.8) | 6 (12.8) | 0 (0) | |
| | Student | 28 (46.7) | 9 (16.1) | 6 (12.8) | 5 (8.1) | |
| | Retired | 0 (0) | 1 (1.8) | 0 (0) | 1 (1.6) | |
| Educational level (%) | Without study | 0 (0) | 0 (0) | 0 (0) | 0 (0) | 0.006 |
| | Primary study | 1 (1.7) | 10 (17.9) | 8 (17) | 3 (4.8) | |
| | Secondary study | 18 (30) | 13 (23.2) | 16 (34) | 12 (19.4) | |
| | Higher studies | 41 (68.3) | 33 (58.9) | 23 (49) | 47 (75.8) | |
| Civil status (%) | Single | 28 (46.7) | 18 (33.1) | 21 (44.7) | 26 (41.9) | 0.081 |
| | Married | 15 (25) | 24 (42.9) | 22 (46.8) | 26 (41.9) | |
| | Widower | 0 (0) | 0 (0) | 0 (0) | 1 (1.6) | |
| | Couple | 17 (28.3) | 14 (25) | 4 (8.5) | 9 (14.5) | |
| Years of training (%) | 1 to 3 years | 9 (15) | 7 (12.5) | 12 (25.5) | 17 (27.4) | 0.018 |
| | 4 to 10 years | 14 (23.3) | 17 (30.4) | 17 (36.2) | 25 (40.3) | |
| | More than 10 years | 37 (61.7) | 32 (57.1) | 18 (38.3) | 20 (32.3) | |
| Federated (%) | Yes | 53 (88.3) | 38 (67.9) | 13 (27.7) | 51 (82.3) | <0.001 |
| | No | 7 (11.7) | 18 (32.1) | 34 (17.2) | 11 (17.7) | |
| Number of sessions (%) | Up to 3 sessions | 11 (18.3) | 11 (19.6) | 19 (40.4) | 4 (6.5) | <0.001 |
| | 4–10 sessions | 40 (66.7) | 45 (80.4) | 27 (57.4) | 49 (75) | |
| | More than 10 sessions | 9 (15) | 0 (0) | 1 (2.1) | 9 (14.5) | |
| Duration of sessions in minutes Mean (SD) | | 93.83 (27.71) | 109.11 (33.64) | 62.66 (20.82) | 86.85 (28.96) | <0.001 |
| Personal trainer (%) | Yes | 45 (75) | 15 (26.8) | 17 (36.2) | 28 (45.2) | <0.001 |
| | No | 15 (25) | 41 (73.2) | 30 (63.8) | 34 (54.8) | |
| | Less than 40% | 3 (5) | 2 (3.6) | 2 (4.3) | 1 (1.6) | 0.072 |
| | 40–59% | 12 (20) | 4 (7.1) | 15 (31.9) | 8 (12.9) | |
| | 60–79% | 33 (55) | 31 (55.4) | 19 (40.4) | 32 (51.6) | |
| | 80–100% | 12 (20) | 19 (33.9) | 11 (23.4) | 21 (33.9) | |

**Table 2.** Descriptive statistics of the dependence on physical exercise (DPE), body disorder or dissatisfaction (BI), motivation and behaviours towards the BI. EDS-R: Exercise Dependence Scale-Revised, BSQ: Body Shape Questionnaire, BREQ-3: Behaviour Regulation in Exercise Questionnaire.

| | | Swimming Mean (SD) n = 60 | Cycling Mean (SD) n = 62 | Athletics Mean (SD) n = 47 | Triathlon Mean (SD) n = 62 | F | *p*-value | Post-hoc Analysis |
|---|---|---|---|---|---|---|---|---|
| Physical exercise dependence. | Withdrawal | 8.57 (3.98) | 8.43 (3.72) | 8.08 (3.69) | 7.98 (3.86) | 0.305 | 0.822 | |
| | Continuance | 7.38 (3.17) | 6.79 (3.34) | 6.81 (3.35) | 7.03 (3.64) | 0.381 | 0.767 | |
| | Tolerance | 10.03 (3.52) | 11.09 (3.96) | 9.97 (4.25) | 10.40 (4.50) | 0.918 | 0.433 | |
| | Lack of control | 7.50 (3.47) | 8.14 (3.56) | 7.38 (3.73) | 7.82 (3.37) | 0.509 | 0.677 | |
| | Reduction of activities | 6.45 (2.73) | 6.64 (2.87) | 5.83 (2.88) | 6.45 (2.87) | 0.775 | 0.509 | |
| | Exercise time | 10.03 (3.99) | 10.63 (4.07) | 8.87 (3.57) | 11.95 (3.32) | 6.374 | <0.001 | T > A ** T > S * |
| | Intention effects | 7.15 (3.46) | 7.64 (3.50) | 6.81 (3.81) | 6.94 (2.66) | 0.652 | 0.583 | |
| | Total EDS-R | 57.12 (16.93) | 59.36 (19.17) | 53.77 (20.13) | 58.58 (16.49) | 0.943 | 0.421 | |
| Body dissatisfaction | Total BSQ | 66.60 (28.10) | 55.84 (21.13) | 58.38 (23.58) | 58.97 (23.35) | 2.260 | 0.082 | |
| Motivation | Intrinsic regulation | 3.42 (0.59) | 3.72 (0.35) | 3.47 (0.54) | 3.63 (0.42) | 4.766 | 0.002 | C > A *. C > S ** |
| | Integrated regulation | 3.50 (0.57) | 3.69 (0.47) | 3.59 (0.47) | 3.59 (0.56) | 3.222 | 0.027 | T > S * |
| | Identified regulation | 3.77 (0.52) | 3.66 (0.47) | 3.59 (0.47) | 3.59 (0.56) | 3.662 | 0.013 | C > S * |
| | Introjected regulation | 1.32 (0.82) | 1.29 (0.81) | 1.19 (0.92) | 1.27 (0.87) | 0.227 | 0.878 | |
| | External regulation | 0.34 (0.53) | 0.21 (0.52) | 0.20 (0.47) | 0.09 (0.24) | 2.987 | 0.009 | S > T * |
| | Amotivation | 0.42 (0.55) | 0.26 (0.58) | 0.19 (0.51) | 0.23 (0.46) | 1.960 | 0.121 | |
| Avoidance behaviours related to body image | Clothing | 1.75 (0.46) | 1.59 (0.47) | 1.70 (0.54) | 1.56 (0.42) | 2.171 | 0.092 | |
| | Social activities | 1.13 (0.42) | 1.14 (0.44) | 1.19 (0.46) | 1.09 (0.28) | 0.601 | 0.615 | |
| | Eating restraint | 1.81 (0.77) | 1.85 (0.75) | 1.78 (0.74) | 1.85 (0.57) | 0.140 | 0.936 | |
| | Checking behaviour/grooming and weighing | 2.59 (0.64) | 2.65 (0.78) | 2.46 (0.74) | 2.46 (0.57) | 1.114 | 0.344 | |
| | Total BIAQ | 1.85 (0.38) | 1.77 (0.43) | 1.79 (0.46) | 1.74 (0.34) | 0.906 | 0.439 | |

Table 3 shows the correlations between the values of DPE, BI, motivation, BIAQ, BMI, age, federated status, and average duration of the training sessions. DPE and the BI showed significant correlations ($p < 0.01$). On the one hand, the DPE was significantly correlated with age ($p < 0.01$), duration of sessions ($p < 0.01$), number of sessions ($p < 0.01$), integrated regulation ($p < 0.05$), introjected regulation ($p < 0.01$), external regulation ($p < 0.01$), demotivation ($p < 0.01$) and tendency of behaviour towards the avoidance of body image (BIAQ; $p < 0.01$). In addition, it was correlated with the dimensions of the BIAQ, including the way of wearing clothes ($p < 0.01$), attitudes to social activities ($p < 0.01$), food restriction ($p < 0.01$) and checking behaviours ($p < 0.01$). BI was shown to correlate significantly with sex ($p < 0.01$), intrinsic regulation ($p < 0.05$), introjected regulation ($p < 0.01$), external regulation ($p < 0.01$), demotivation ($p < 0.01$), and tendency of behaviours towards the avoidance of body image (BIAQ; $p < 0.01$). In addition, it correlated with the dimensions of the BIAQ, including the way of wearing clothes ($p < 0.01$), attitudes to social activities ($p < 0.01$), food restriction ($p < 0.01$) and checking behaviours ($p < 0.01$).

By performing the same correlations but segmented by sex, the following results were obtained. For the male gender, EDS-R and BSQ scores were significantly correlated (r = 0.312; $p < 0.01$). In men, DPE and BI were correlated (r = 0.312; $p < 0.01$). DPE was correlated with the number of sessions completed (r = 0.212; $p < 0.05$), age (r = –0.293; $p < 0.01$), introjected regulation (r = 0.514; $p < 0.01$), external regulation (r = 0.289; $p < 0.01$) and the tendency toward negative conducts related to body image (r = 0.351; $p < 0.01$). In relation to the dimensions of the BIAQ, it was correlated with the way of wearing clothes (r = 0.347; $p < 0.01$), attitudes to social activities (r = 0.2301; $p < 0.01$), food restriction (r = 0.201; $p < 0.01$) and checking behaviours (r = 0.265; $p < 0.01$). The IC was correlated with introjected regulation (r = 0.510; $p < 0.01$), external regulation (r =0. 489; $p < 0.01$), demotivation (r = 0.214; $p < 0.01$), BMI (r = 0.393; $p < 0.01$) and the tendency of behaviours related to body image (r = 553; $p < 0.01$). BI was also correlated with the dimensions of the BIAQ including the mode of wearing clothes (r = 0.429; $p < 0.01$), attitudes to social activities (r = 0.333; $p < 0.01$), food restriction (r = 0.481; $p < 0.01$) and checking behaviours (r = 0.413; $p < 0.01$). In females, DPE was also significantly correlated with BI (r = 0.404; $p < 0.01$). On the one hand, BI was correlated with intrinsic regulation (r = –0.300; $p < 0.01$), introjected regulation (r = 0.252; $p < 0.05$), demotivation (r = 0.254; $p < 0.05$) and the tendency to have negative conducts in relation to body image (r = 0.601; $p < 0.01$). In addition, it was correlated with the dimensions of the BIAQ including the way of wearing clothes (r = 0.519; $p < 0.01$), attitudes to social activities (r = 0.242; $p < 0.01$), food restriction (r = 0.477; $p < 0.01$) and checking behaviours (r = 0.513; $p < 0.01$). DPE was correlated with the number of sessions (r = 0.386; $p < 0.01$), age (r = –0.311; $p < 0.01$), duration of sessions (r = 0.450; $p < 0.01$) introjected regulation (r = 0.440; $p < 0.01$) and the tendency to have negative conducts related to body image (BIAQ; r = 0.509; $p < 0.01$). In addition, it was correlated with the dimensions of the BIAQ including the way of wearing clothes (r = 0.385; $p < 0.01$), attitudes to social activities (r = 0.377; $p < 0.01$), food restriction (r = 0.428; $p < 0.01$) and checking behaviours (r = 0.316; $p < 0.01$).

Tables 4 and 5 show the linear regression of the factors that predict BI and DPE, respectively. Common predictive factors were the introjected regulation and the tendency to have negative conducts related to body image.

**Table 3.** Pearson correlations between the DPE, BI, motivation, age, body mass index (BMI), duration of training and behaviours related to the avoidance of BI.

| | A | B | C | D | E | F | G | H | I | J | K | L | M | N |
|---|---|---|---|---|---|---|---|---|---|---|---|---|---|---|
| Number of sessions (A) | 1 | 0.211 (**) | −0.297 (**) | 0.076 | 0.106 | 0.188 (**) | −0.084 | 0.076 | 0.041 | 0.065 | 0.069 | 0.272 (**) | −0.114 | 0.057 |
| Duration of sessions (B) | | 1 | −0.213 (**) | 0.028 | 0.115 | 0.059 | 0.054 | 0.059 | 0.009 | −0.009 | 0.019 | 0.252 (**) | −0.112 | 0.059 |
| Age (C) | | | 1 | −0.126 | 0.039 | 0.018 | 0.183 (**) | −0.134 (*) | −0.137 (*) | −0.070 | −0.129 | −0.304 (**) | 0.284 (**) | −0.144 (*) |
| Gender (D) | | | | 1 | 0.048 | 0.097 | 0.071 | 0.032 | −0.042 | −0.011 | 0.339 (**) | 0.055 | −0.426 (**) | 0.246 (**) |
| Intrinsic regulation (E) | | | | | 1 | 0.433 (**) | 0.344 (**) | −0.090 | −0.314 (**) | −0.167 (*) | −0.157 (*) | 0.063 | −0.082 | −0.204 (**) |
| Integrated regulation (F) | | | | | | 1 | 0.374 (**) | 0.014 | −0.245 (**) | −0.205 (**) | −0.017 | 0.158 (*) | −0.112 | −0.096 |
| Identified regulation (G) | | | | | | | 1 | 0.021 | −0.124 | −0.220 (**) | −0.007 | 0.012 | −0.016 | −0.113 |
| Introjected regulation (H) | | | | | | | | 1 | 0.419 (**) | 0.186 (**) | 0.396 (**) | 0.484 (**) | 0.098 | 0.303 (**) |
| External regulation (I) | | | | | | | | | 1 | 0.480 (**) | 0.303 (**) | 0.206 (**) | 0.013 | 0.327 (**) |
| Amotivation (J) | | | | | | | | | | 1 | 0.211 (**) | 0.166 (*) | 0.027 | 0.253 (**) |
| Total BSQ (K) | | | | | | | | | | | 1 | 0.349 (**) | 0.118 | 0.603 (**) |
| Total EDS-R (L) | | | | | | | | | | | | 1 | 0.014 | 0.412 (**) |
| BMI (M) | | | | | | | | | | | | | 1 | 0.091 |
| Total BIAQ (N) | | | | | | | | | | | | | | 1 |

**Table 4.** Linear regression among the predictive factors of BI.

| | Non-Standardized Coefficient | | Standardized Coefficient | T | *p*-Value | 95% Confidence Interval for B | |
| --- | --- | --- | --- | --- | --- | --- | --- |
| | B | Standard Error | Beta | | | B | Standard Error |
| (Constant) | −9.231 | 11.513 | | −0.802 | 0.424 | −31.925 | 13.462 |
| Gender | 13.243 | 2.916 | 0.259 | 4.542 | 0.000 | 7.495 | 18.992 |
| Intrinsic regulation | −1.669 | 2.643 | −0.034 | −0.632 | 0.528 | −6.878 | 3.540 |
| Introjected regulation | 5.809 | 1.753 | 0.201 | 3.314 | 0.001 | 2.354 | 9.264 |
| External regulation | 3.655 | 3.398 | 0.068 | 1.076 | 0.283 | −3.043 | 10.352 |
| Amotivation | 2.006 | 2.629 | 0.043 | 0.763 | 0.446 | −3.176 | 7.188 |
| PED | 0.051 | 0.083 | 0.038 | 0.622 | 0.534 | −0.111 | 0.214 |
| Clothing | 6.459 | 8.006 | 0.124 | 0.807 | 0.421 | −9.321 | 22.240 |
| Social activities | −1.023 | 4.983 | −0.017 | −0.205 | 0.837 | −10.845 | 8.798 |
| Eating restraint | 8.853 | 3.527 | 0.254 | 2.510 | 0.013 | 1.900 | 15.805 |
| Checking behaviour/grooming and weighing | 7.115 | 3.406 | 0.198 | 2.089 | 0.038 | 0.401 | 13.830 |
| Total BIAQ | 1.019 | 16.871 | 0.017 | 0.060 | 0.952 | 32.237 | 34.275 |
| R2 | 0.497 | 0.471 | | | | | |

**Table 5.** Linear regression of the predictive factors of DPE.

| | Non-Standardized Coefficient | | Standardized Coefficient | T | *p*-Value | 95% Confidence Interval for B | |
| --- | --- | --- | --- | --- | --- | --- | --- |
| | B | Standard Error | Beta | | | B | Standard Error |
| (Constant) | 1.349 | 13.775 | | 0.098 | 0.922 | −25.806 | 28.503 |
| Number of sessions | 4.031 | 1.940 | 0.116 | 2.078 | 0.039 | 0.208 | 7.855 |
| Duration of sessions (mean) | 0.085 | 0.030 | 0.152 | 2.849 | 0.005 | 0.026 | 0.143 |
| Age | −0.328 | 0.101 | −0.190 | −3.243 | 0.001 | −0.527 | −0.129 |
| Integrated regulation | 4.888 | 2.052 | 0.132 | 2.383 | 0.018 | 0.844 | 8.933 |
| Introjected regulation | 7.919 | 1.275 | 0.372 | 6.209 | 0.000 | 5.405 | 10.433 |
| External regulation | −3.688 | 2.589 | −0.094 | −1.425 | 0.156 | −8.791 | 1.415 |
| Amotivation | 1.199 | 2.037 | 0.035 | 0.589 | 0.557 | −2.816 | 5.214 |
| BMI | 0.153 | 0.417 | 0.021 | 0.368 | 0.714 | −0.669 | 0.976 |
| Clothing | 8.750 | 5.796 | 0.229 | 1.510 | 0.133 | −2.676 | 20.177 |
| Social activities | 10.864 | 3.584 | 0.240 | 3.031 | 0.003 | 3.799 | 17.928 |
| Eating restraint | 5.661 | 2.603 | 0.220 | 2.175 | 0.031 | 0.530 | 10.793 |
| Checking behaviour/grooming and weighing | 2.519 | 2.453 | 0.095 | 1.027 | 0.306 | −2.317 | 7.355 |
| Total BIAQ | −13.194 | 11.732 | −0.293 | −1.125 | 0.262 | −36.321 | 9.932 |
| Total BSQ | 0.025 | 0.050 | 0.033 | 0.488 | 0.626 | −0.075 | 0.124 |
| R2 | 0.463 | 0.427 | | | | | |

## 4. Discussion

First of all, we have to mention the scarce literature found on the relationships among our different variables. In addition, the variety of instruments used for previous studies as well as the conceptual problem with defining addiction to sport make the results of the different studies difficult to compare. Therefore, we must be cautious in our discussion of other investigations in relation to our study.

The first objective was to study the sociodemographic differences that could exist among the different studied individual sports. Few studies have investigated this area; however, Latorre et al. [24] agreed that triathletes are the most likely to be federated and perform the most sessions in a week, although our study revealed similar percentages for swimmers in both cases. Likewise, the Latorre et al. study showed similar results for cyclists in terms of the duration of sessions. Similarly, in their study, significant differences in BMI were shown, with runners having the lowest BMI. Finally, there were also differences in the sample as a whole and in all sports, except in swimming, with respect to BMI, which was higher in men than in women, as normally, women are smaller than men.

The second objective of this study was to analyse DPE and BI. Sussman et al. [35] observed that the prevalence of DPE in the general population of the USA to be between 3% and 5%. In a study on university students, Reche et al. [36] found a DPE prevalence of 8% in athletes participating in individual sports. For Harris et al. [37], it was between 16% and 36% depending on which degree they studied. Latorre et al. [24] showed that, in individual sports, there is a DPE prevalence of 13.6%, with significant differences among different sports. Specifically, triathlon was shown to have the highest prevalence with 29.6% as opposed to our study where swimmers (10%) were shown to have the highest DPE prevalence. Magee, Buchanan and Barrie [38] and Valenzuela and Arriba-Palomero [39] obtained

an RD of 8.6% in a study with male triathletes, while Blaydon and Linder (2002) [13] analysed triathletes and found that 25%–30% showed symptoms of sport addiction. Similarly, Youngman and Simpson [40] found that approximately 20% of their sample of triathletes were at risk of sport addiction.

Taking into account the comparison of sexes, several studies have shown conflicting information. Ruiz-Juan, Zarauz and Flores-Allende [41] studied the negative addiction to running (ANC) in endurance runners (half marathon and marathon), finding an average score higher than the mean without significant differences between sexes. Ortiz and Arbinaga [42] and González-Cutre and Sicilia [17] in studies on individual sports and conditioning centres did not find significant differences in DPE prevalence in relation to sex. Modolo et al. [20] found that 28% of women and 38% of men had DPE symptoms without finding significant differences between genders. Bingol and Bayansalduz [43] concluded that 19.2% of women and 13.15% of men suffered from RD in a multitude of studied sports. Mayolas-Pi et al. [44], in a study with cyclists (females and males), showed that 17% of men and 16% of women were at risk of sport addiction. We found a few studies with significant differences in DPE prevalence, showing higher results in women than that shown in our results.

In the study of DPE and its dimensions compared to each sport modality, Latorre et al. [24] found differences in the total score, which was higher in triathletes and cyclists than in runners, a fact that was not observed in our study. However, there were similarities in the results of the time dimension of exercise, which was found to be higher in triathletes than in athletics competitors, although in our study, triathletes also exercised more than swimmers. Likewise, Harris et al. [37], conducted a study with university students where differences were found in the subscale "exercise time" among students of Sports Science and those enrolled in other university degrees.

Like in our research, Ortiz and Arbinaga (2016) [42] did not find significant differences in the subscales of the EDS-R between the two sexes. However, Gonzalez-Cutre and Sicilia [17], in a study on fitness centres, found significant differences between men and women in all subscales of DPE, with scores being higher in the male sex, except for the abstinence factor. It is important to note that the sample in Ortiz and Arbinaga [42] was much smaller than ours.

As far as BI and its relation to sport, in this case individual sports, is concerned, Fortes, Almeida and Ferreira [45] showed that almost 15% of their sample suffered BI tendencies. Their sample included athletes participating in various sport modalities. This result is inferior to this study, and even more so in comparison to swimmers. Latorre et al. [24], found a higher prevalence of BI of 16.3%, with swimmers having the highest BI with 22.6%, obtaining similar results to those shown in our study. Regarding the study of BI among athletes based on gender, many studies have confirmed that women suffer from greater BI than men (Zanon et al. [46]; Zmijewski and Howard [19]). This is supported in our study by the results for the conduct of body image avoidance (BIAQ).

In terms of the association between these two variables, a significant relationship was shown, as can be observed in the results of other studies where a strong association between DPE and BI has been shown (Davis [47], Cook and Hausenblas [14], Weinstein and Weinstein [15]). Zmijewski and Howard [19]), in a study with Physical Activity and Sports Science university students, concluded that girls with higher DPE scores showed higher BI scores, while in boys this did not happen, confirming the significant differences that we found in our study regarding the greater BI scores in women than in men. Hausenblas and Fallen (2002) [7] showed that, at a general level, the subscale of lack of control predicted the BI score. These same authors found that, in men, the intention subscale and time of exercise were predictors of BI.

The next objective was to establish the relationship between the DPE and the BI in relation to motivation, in particular the regulation conducts of motivation.

In this regard, several studies have indicated that introjected regulation is a major predictor of DPE (Edmunds, Ntoumanis and Duda [48], Fortier and Farrell [49]). On the other hand, González-Cutre and Sicilia [50] established in their study that not only is DPE related to non-self-determined motivations such as external and introjected regulation, but it is also linked to some self-determined motivation factors such as integrated regulation. Our study coincides with these results, adding demotivation as

a factor that prevails in DPE. These authors conclude that the positive or negative characteristics of high DPE could be determined by how the person constitutes their behaviours within their lifestyle. Latorre et al. [24] found that integrated, identified, and introjected regulation were associated with DPE.

Zarauz, Ruiz-Juan and Arbinaga [51] studied sport addiction and the running compromise (EAG) in athletes and found that high scores of females in two self-determined motivation indicators (psychological and life—self-esteem objectives) predicted the presence of EAG. However, in males, EAG could be significantly predicted by high scores in the three more self-determined motivation factors and lower scores in the non-self-determined motivation factors.

Sicilia et al. [52] indicated that, in a multitude of occasions, the practice of exercise is preceded by external or internalised pressure that motivates an individual to do exercise to lose body weight and maintain a good body image. Regarding this, Latorre et al. [24], stated that only introjected regulation is associated with BI. On the other hand, in our study, the three types of non-self-determined regulation (introjected, external and demotivation) were negatively associated with intrinsic regulation.

However, although physical and health motivations can be associated with the positive consequences of physical exercise for people who have low body dissatisfaction, greater support for the motivations of both physical fitness and health, like appearance and weight, are associated with higher dissatisfaction of body state in women, categorised as a highly dissatisfied body image (Sampasa-Kanyinga et al. [53]).

Gonzalez-Cutre and Sicilia [17] included integrated regulation and external regulation as predictors of DPE, in contrast to this study, which only identified integrated regulation as a predictor. Latorre et al. [24] identified regulation as predicting DEP. Regarding other variables, a multitude of studies and instruments have shown that the prevalence of sport addiction decreases with age, so it correlates inversely (Szabo [54]; Ruiz-Juan, Zarauz y Arbinaga [55]; Costa et al. [56]). In addition, in our study, the number of sessions and their duration (Reche et al. [36]; Reche and Gomez [57]) were found to be related to DPE.

Some studies have made reference not only to the training or the duration of training itself but also to the fact that there are differences in relation to the tests that are carried out (Guszkowska and Rudnicki [25]). Preparation for a sprint triathlon is not the same as for an ironman triathlon (Youngman and Simpson [40]), and preparation for a half marathon is not the same as for a marathon (Ruiz-Juan et al. (2016)). In terms of predictors of DPE, an inversely significant relationship with age was found (Latorre et al. [24]). Social activities were found to be one of the conducts that prevails in DPE. As for BI, in addition to the previously mentioned factors, sex was shown to be associated with BI. This coincides with our study, where sex was shown to be an important factor in BI (Harris et al. [37]; Hausenblas and Fallon [7]).

To determine which variables or factors predict the occurrence of DPE and BI, a multiple linear regression was conducted. Introjected regulation was shown to predict the two, as was shown in other studies (Latorre et al. [24]). In the analysis of data, a comparison was carried out using the conducts of avoidance, with food restriction being used as a predictor for both variables.

The biggest difficulty for this study was finding women who practiced or competed in cycling. In addition to this, the data collection process was underestimated, since I thought it would be easier to perform. Another important aspect to take into account in the sociodemographic analysis is not only the occupations of participants but also the number of working hours per week as well as the family burden, since having children or not is an important factor. Another important limitation is the great distances and kilometres covered to gain a sufficient sample, which was self-funded and included a large investment in time. Finally, and in relation to the questionnaires, the extension of them was highlighted, especially in the BSQ questionnaire where most of the participants issued a complaint of being too long. This led some questionnaires being eliminated due to not being completed.

## 5. Conclusions

In view of our results, we observed that 8.5% of participants were classified as RD, but those classified as AND reached 48.4%. In many cases, participants had a score higher than 5 in two dimensions or a total score of EDS-R that was much higher than the average. This suggests that many of the participants could be in danger of DR, without little difference shown between women and men. In our study, the prevalence of RD was greater in women, particularly in swimmers where 1 out of 4 were classified as having RD.

In addition, we can conclude that the analysed individual athletes did not show a great prevalence of BI, but we must take into account the difference between BI in men and women, a fact that many studies have corroborated in all types of people (athletes and non-athletes).

In our study, it was found that there are several validated instruments for the DPE that do not give the same results or for which results may differ among researchers, which indicates that addiction manifests itself in different ways and the context in which the investigation is carried out must be taken into account.

Finally, we observed that DPE and BI are correlated, and both can be preceded by a high level of motivation that is not self-determined, highlighting intrinsic regulation as a predictor of both parameters. Together with this regulation, food restriction was the other predictor variable found between the two variables. A scarce amount of literature exists on these conducts of avoidance in relation to the DPE.

Our results, together with previous studies that have been done in this area, show how we can identify the profiles of endurance athletes in order to create strategies to prevent and treat (if necessary) addiction in sports.

**Author Contributions:** Conceptualization, I.T.-Q. and J.J.G.F.; Methodology, E.C.V. and Á.S.R.; Investigation, I.T.-Q.; Writing—Original draft Preparation, I.T.-Q. and J.S.-P.; Writing—review and editing, E.C.V.; Supervision, J.S.-P. and Á.S.R.

**Funding:** This research received no external funding.

**Acknowledgments:** The authors would like to thank the teachers and pupils who participated in this study.

**Conflicts of Interest:** The authors declare no conflict of interest.

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
