# Peer review of "Risk of Dependence on Sport in Relation to Body Dissatisfaction and Motivation"

_sustainability, doi:10.3390/su11195299_

Round 1

Reviewer 1 Report

Thank you for let me review this article.

Here are some comments

First of all, I think this article better fit the aim and scope of  IJERPH, so I suggest you to submit to this journal.

Below are some suggestions to this article

First of all, not all abbreviations were introduced in the article. For example, what is CI? what is HF?

Abstract:

the contributions of this article were not well addressed.

the regression results indicated that introjected regulation eating restraint, checking behavior could predict HF. Also, number and Duration of sessions, age, integrated, introjected, external regulations social activities, eating restraint could all predicted DPE. It was not all mentioned in the abstract.

Introduction

How you link variables to DPE and BI were not well addressed.

Instruments

not well-introduced. For example, how you identify who were at risk of dependence?

Isn't it better to present your questionnaire?

Results

the format of tables were not adequate

For some sample size under 5, it is not appropriate to use T-test or ANOVA. And How you get the SD if you got only 1 sample?(Table 2, swimming education level= primary school}

Also comparing the difference of BMI made no sense since it is only a index categorized for body composition. What is it for to compare this variable?

discussion

Can you explain why in swimming, why women's BMI is lower than men?(P.11 Ln#262)

Ln#273 Can you explain why?

Ln#355-367 I suggest you move this section to limitation section

Conclusion

I suggest to make a shorter description to make your contributions clearer. 

Author Response

Open Review

English language and style

( ) Extensive editing of English language and style required  
( ) Moderate English changes required  
(x) English language and style are fine/minor spell check required  
( ) I don't feel qualified to judge about the English language and style  

Yes

Can be improved

Must be improved

Not applicable

Does the introduction provide sufficient background and include all relevant references?

( )

( )

(x)

( )

Is the research design appropriate?

( )

( )

(x)

( )

Are the methods adequately described?

( )

( )

(x)

( )

Are the results clearly presented?

( )

( )

(x)

( )

Are the conclusions supported by the results?

( )

( )

(x)

( )

Comments and Suggestions for Authors

Thank you for let me review this article.

Here are some comments

First of all, I think this article better fit the aim and scope of  IJERPH, so I suggest you to submit to this journal.

Below are some suggestions to this article

First of all, not all abbreviations were introduced in the article. For example, what is CI? what is HF?

CHANGE MADE.

Abstract:

the contributions of this article were not well addressed.

the regression results indicated that introjected regulation eating restraint, checking behavior could predict HF. Also, number and Duration of sessions, age, integrated, introjected, external regulations social activities, eating restraint could all predicted DPE. It was not all mentioned in the abstract.

CHANGE MADE.

Introduction

How you link variables to DPE and BI were not well addressed.

IT CAN NOT BE CHANGED.

Instruments

not well-introduced. For example, how you identify who were at risk of dependence?

IT IS EXPLAINED ON THE LINES 104-108.

Isn't it better to present your questionnaire?

THE FORMAT WAS NOT SUITABLE.

Results

the format of tables were not adequate

For some sample size under 5, it is not appropriate to use T-test or ANOVA. And How you get the SD if you got only 1 sample?(Table 2, swimming education level= primary school}

THERE IS NO SAMPLE OF 5 SUBJECTS

Also comparing the difference of BMI made no sense since it is only a index categorized for body composition. What is it for to compare this variable?

CHANGE MADE

Discussion

Can you explain why in swimming, why women's BMI is lower than men?(P.11 Ln#262)

CHANGE MADE

Ln#273 Can you explain why?

I DONT KNOW.

Ln#355-367 I suggest you move this section to limitation section.

LIMITATIONS HAVE BEEN INCLUDED IN THE DISCUSSION SECTION

Conclusion

I suggest to make a shorter description to make your contributions clearer. 

CHANGE MADE.

PLEASE SEE THE ATTACHMENT WITH CHANGE

Reviewer 2 Report

Overall, the text needs to be edited for understanding. The tables need to be reformatted for clarity. 

A factor analysis needs to be done for the BREQ-3, the Cronbach alpha is below the acceptable 0.8.

For the linear regression, there are many potential confounding variables. The authors need to control for all the baseline differences (age, employment, education level, years of training, etc...).

I must assert that there needs to be extensive editing to this manuscript. The language use is quite poor. Abstract: Alter the phrase, “getting to know” this is not scientific verbiage.  “The obtained results” is redundant, please revise. “Tend to have corporal dissatisfaction” do they or don’t they? This is what you measured. Overall, the abstract is fine, it just needs minor language/word choice editing. Introduction: “Nowadays” should not be used to begin a scientific article. And psychic? Do the authors mean psychological? Line 33: conjectures? Do the authors mean connotations?  Line 37: The transition into this paragraph didn’t make sense. Perhaps…. Due to a need in the field Ogden et al., operationalized the definition of DPE…? The gap or need for more research is not well supported. Clarify what past research has done and how this investigation fills a gap. Methods: The authors did a good job of explaining the different questionnaires. However, the low Cronbach alpha for the BREQ-3 needs to be addressed or exclude this data entirely. Perhaps the authors could use a factor analysis to see what potential constructs arise with their sample. Line 142: Omit “First of all” this phrase should not show up in a statistical analysis section. Results: There are many spacing problems in Table 1- BMI, Years of training (%), Duration of sessions in minutes… Please fix these so that the reader call follow along. Table 3 needs to be altered for ease of reading. The lettering helps, but spacing and abbreviations would likely be better. Table 4- Gender was used in the linear regression, this is reasonable, but the authors need to use ALL the sociodemographic characteristics in this regression. Gender predicted the most variance for HF. DPE was similar, the authors need to control for more of the demographic data. Discussion: Omit, “First of all”. Line 251- Change “that are used” to “that were used” Line 252- reword “making the results of the different studies differ among them” Line 369-372 please provide a citation. Line 388- correlate with each other. Overall, this is a very interesting study; however, there is a need for more editing to make the paper clear to the readers of the journal. The only issue methodically is the lack of using more sociodemographic variables for the linear regressions, which seem like they could explain more variance than the questionnaires utilized.

Author Response

Open Review

English language and style

(x) Extensive editing of English language and style required  
( ) Moderate English changes required  
( ) English language and style are fine/minor spell check required  
( ) I don't feel qualified to judge about the English language and style  

Yes

Can be improved

Must be improved

Not applicable

Does the introduction provide sufficient background and include all relevant references?

( )

( )

(x)

( )

Is the research design appropriate?

( )

(x)

( )

( )

Are the methods adequately described?

( )

( )

(x)

( )

Are the results clearly presented?

( )

( )

(x)

( )

Are the conclusions supported by the results?

( )

( )

(x)

( )

Comments and Suggestions for Authors

Overall, the text needs to be edited for understanding. The tables need to be reformatted for clarity. 

A factor analysis needs to be done for the BREQ-3, the Cronbach alpha is below the acceptable 0.8.

According to Celina and Campo, 2005, the acceptable value would be 0.7. We have a value of 0.642

For the linear regression, there are many potential confounding variables. The authors need to control for all the baseline differences (age, employment, education level, years of training, etc...).

It could be done ... but it would be another analysis

Please see the attachment with change

Round 2

Reviewer 1 Report

The format of tables needs to be improved. Especially, the decimal point should replace comma.

Author Response

Good morning dear reviewer, we appreciate the corrections as they have helped us to improve the initial document.

In this new version the small changes requested have been made and we have also passed the English revision carried out by the MPDI platform.

thank you

Jesús Sáez

Reviewer 2 Report

Looks good, thank you for the edits.

Author Response

Good morning dear reviewer, we appreciate the corrections as they have helped us to improve the initial document.

In this new version the small changes requested have been made and we have also passed the English revision carried out by the MPDI platform.

thank you
